# Long-Term Outcomes after Multimodal Treatment for Clival Chordoma: Efficacy of the Endonasal Transclival Approach with Early Adjuvant Radiation Therapy

**DOI:** 10.3390/jcm12134460

**Published:** 2023-07-03

**Authors:** Hyun Dong Yoo, Jong Chul Chung, Ki Seok Park, Seung Young Chung, Moon Sun Park, Seungjun Ryu, Seong Min Kim

**Affiliations:** 1Department of Neurosurgery, Eulji University Hospital, Eulji University College of Medicine, Daejeon 35233, Republic of Korea; 20190107@eulji.ac.kr (H.D.Y.); ks3432@eulji.ac.kr (K.S.P.); neurocsy@eulji.ac.kr (S.Y.C.); umspark@eulji.ac.kr (M.S.P.); 2Center for Neuromodulation, Department of Neurosurgery, NYU Langone Medical Center, New York, NY 11021, USA; jcwithjc7@gmail.com

**Keywords:** clival chorodma, endonasal transclival approach, skull base tumor, adjuvant radiation therapy

## Abstract

This study investigates the long-term outcomes of clival chordoma patients treated with the endonasal transclival approach (ETCA) and early adjuvant radiation therapy. A retrospective review of 17 patients (2002–2013) showed a 10-year progression-free survival (PFS) rate of 67.4%, with the ETCA group showing fewer progressions and cranial neuropathies than those treated with combined approaches. The ETCA, a minimally invasive technique, provided a similar extent of resection compared to conventional skull-base approaches and enabled safe delivery of high-dose adjuvant radiotherapy. The findings suggest that ETCA is an effective treatment for centrally located clival chordomas.

## 1. Introduction

Clival chordomas are invasive cranial bone tumors that arise from ectopic notochord remnants at the central skull base with an incidence of 0.08–0.1/100,000 [1,2,3,4,5,6]. These tumors are anatomically based on the clival bone and infiltrate and destroy adjacent areas, such as the sphenoid and nasopharynx anteriorly; the petrous apex, Meckel’s cave, and pons posterolaterally; the parasellar, including the cavernous sinus, superiorly; and the upper cervical bone and larynx inferiorly. Because of their local invasiveness and rates of recurrence, clival chordomas are clinically malignant and have a relatively poor survival prognosis of <12 months if the patient is left untreated [5,7].

Maximum surgical resection is the most basic requirement for treating clival chordomas [4,8,9,10,11]. These tumors are good candidates for endoscopic approaches because of their predominantly midline extradural origin and growth, soft consistency, and minimal hemorrhagic characteristics [7,11,12,13,14,15,16,17]. The endonasal transclival approach (ETCA) is applied according to the specific location and the craniocaudal spread of the clival chordoma to the central skull base. However, gaining access to the paraclival area requires a combination of extended lateral, posterolateral, and anterior skull-base approaches utilizing multiple corridors. The combined approaches include a mix of techniques utilized based on the location of the lateral invasion, including, but not limited to, the extradural transcavernous approach, the far-lateral transcondylar approach, and the anterior petrosal approach. Therefore, extensive resection of a complex paraclival pathology is achieved by combining endoscopic surgery and conventional skull-base approaches.

However, radical resection remains difficult and often results in higher rates of surgical morbidity. Adjuvant radiotherapy for residual or recurrent tumors is also recommended for long-term tumor control [18,19,20,21,22]. Adjuvant radiation to remnant lesions is restricted in some cases, such as when the tumor invades the brainstem dura mater or arachnoid membrane, or when the tumor contacts cranial nerves. The ETCA can reduce sufficient tumor burden of such critical structures with less morbidity than that of other approaches. Furthermore, the ETCA contributes to functional outcomes during long-term management of patients with clival chordomas.

There are a limited number of cases and a lack of long-term clinical outcomes data in the literature regarding patients with chordomas who were treated using the ETCA. The purpose of this study was to reconfirm the role of the ETCA as the first surgical step for multimodal treatment of clival chordomas. We analyzed clinical and surgical factors statistically and investigated the details of the surgical techniques to achieve better long-term tumor control.

## 2. Materials and Methods

### 2.1. Patient Population and Treatment Modalities

A total of 17 patients who had been surgically treated for clival chordomas between 2002 and 2013 were eligible for this study. Tumor volume, location, extent of resection, number of operations, number of previous surgeries or radiation therapy, use of adjuvant radiation therapy, and clinical status were analyzed.

All patients who were suspected of a remnant tumor received postoperative adjuvant radiation therapy. Six patients received proton beam radiation therapy (PBRT) and four patients received GKRS, while six patients who achieved gross total resection (GTR) were not scheduled for adjuvant therapy. Postoperative magnetic resonance imaging (MRI) was obtained at 3, 6, and 12 months and annually after 1 year unless the patient showed symptoms. All the information mentioned above is summarized in Table 1.

### 2.2. Surgical Strategy

Surgical procedures were adopted based on tumor location and extension. A neuronavigation system based on preoperative MRI was used in all patients. We divided the cases into four classifications based on surgical strategy level: Level I: extradural lesion with or without minimal dural contact, in which the tumor was limited to the clival bone and prepontine area; Level II: moderate intradural invasion or pure intradural lesion that compressed the brainstem posteriorly and occupied the parasellar area superiorly; Level III: lesion extending into a lateral area, such as the cavernous sinus, Meckel’s cave, cerebellopontine angle cistern, or jugular fossa; and Level IV: lesion extending downward to the upper cervical area; tumor arises from the lower clivus, occupies the pharyngeal space extracranially and the medulla or spinal cord near the foramen magnum (Table 1).

### 2.3. Endonasal Transclival Approach

The monostril endoscopic approach began with an endonasal mucosal incision using a neuroendoscope (0, 30, and 70°rod lenses; Karl Storz, Tuttllingen, Germany). The middle turbinate had deviated to the contralateral side, and a sphenoidotomy was performed through the sphenoid ostium as much as possible. The upper clivus was just posteroinferior to the dorsum sellae and was often eroded by the tumor. An extradural tumor located at the upper clivus was removed easily by drilling the sellar floor and the eroded clival bone. Both 30 and 70°angled endoscopes were introduced to visualize the corner near the carotid artery. The posterior clinoid process was removed behind the pituitary gland, and the bone around the internal carotid artery (ICA) was skeletonized with meticulous drilling using a diamond burr. Tumors adjacent to the brainstem or cranial nerves were grasped and piecemeal resected relatively safely. Wide viewing angles allowed us to visualize every corner of the cavity without blind spots, although we worked through a narrow corridor that permitted only one endoscope and single-shaft microforceps. We approached the posterior nasopharynx along the floor of the palate for a lower clivus or upper cervical lesion. We typically prepared a pedicled mucosal flap before preparing the bone window. Dural repair was supported by fat and fascia lata as an inlay graft and covered by a vascularized mucosal flap for dural defects or for a wide dural opening.

### 2.4. Statistical Analysis

Progression-free survival (PFS) and overall survival (OS) rates were determined using the Kaplan–Meier analysis. Associations between time of progression from initial treatment, extent of tumor resection, surgical approach, and remnant brainstem lesions and PFS were evaluated by univariate analysis using the log-rank test. GTR was defined as a resection margin that included normal bone tissue or dural margin without any tumor cells in a frozen section biopsy, intraoperatively. Near-total resection (NTR) and subtotal resection (STR) were substantial (>90% tumor volume) resection and central debulking (<90% tumor volume resection), respectively. The ETCA was compared with combined lateral skull-base approaches. A remnant brainstem lesion was identified by the surgeon perioperatively and using postoperative MRI. Multivariate analyses were performed using the Cox proportional hazards model to detect factors independently affecting PFS. A *p*-value < 0.05 was considered significant. The SPSS ver. 18 for Windows software (SPSS Inc., Chicago, IL, USA) was used for all statistical analyses.

## 3. Results

Seventeen patients (13 females and 4 males, including 3 pediatric patients; mean age, 38.7 years; range, 8–59 years) were included. The mean follow-up period was 66.7 months (range, 9–132 months; median, 62 months). Five of the seventeen patients had been treated previously at other institutes at the time of this study. Operations had been performed on three patients (two trans-sphenoidal approaches and one suboccipital approach), whereas Gamma Knife radiosurgery (GKRS) had been performed at the institutes on two patients. The patients underwent 24 surgeries, including 18 ETCAs, at our institute during the study period. Eleven patients only underwent the ETCA, and 4 of the 17 patients underwent combined lateral skull-base approaches followed by the ETCA. Three patients underwent extradural trans-cavernous sinus approaches (TCSA) for tumors that extended laterally into the cavernous sinus and Meckel’s cave. One patient underwent an anterior trans-petrosal approach (ATPA) to resect a tumor that extended into the petrous apex, cerebellopontine angle cistern, and jugular fossa. The remaining two patients only underwent ATPA without ventral approach.

### 3.1. Tumor Locations and Invasion of Critical Structures

Tumors in most of the patients arose from the posterior surface of the clivus, with the exception of one case in which the tumor was embedded in the intradural space. All patients had tumors occupying the clivus and prepontine cistern. Four cases (23.5%) were limited to the clivus and extradural space and did not invade the dura mater. Six patients (35.3%) had tumors largely located on the clivus and extradural space but which had invaded the dura and compressed the brainstem or neurovascular structures. We found one case (5.9%) of a pure intradural chordoma invaginating into the brainstem and encasing the basilar artery. Tumors in eight cases (47%) were found in the parasellar area (cavernous sinus or Meckel’s cave), and tumors in five cases (29.4%) extended posterolaterally to the petrous bone and cerebellopontine angle cistern. Two cases (11.8%) had tumors that extended down toward the cervicomedullary junction.

### 3.2. Overall Outcomes

Sixteen of the seventeen (94.1%) patients were alive at the time of the study. Only one patient had poor clinical status and died because she had brainstem dysfunction prior to surgery. This patient (patient 16) showed altered mental status, hemiparesis, and multiple cranial neuropathies. Although she was decompressed immediately using the ATPA, she died after 9 months of a bedridden state. The overall 10-year PFS rate was 67.4%. The 3- and 5-year PFS rates were 87.4% and 78.7%, respectively (Figure 1A). Thirteen of the seventeen patients (76.5%) never progressed, but three patients (17.6%) required additional treatment. Progression was diagnosed at 33, 40, and 49 months after initial treatment. Patient 15 underwent the ATPA for clival chordomas that expanded to the petrous apex and immediate GKRS for a remnant cavernous sinus mass during initial treatment. A tumor regrew at the periphery of the 50% isodose line after 33 months but was stationary for 7 years after additional GKRS. Patient 17, who presented with the largest (98.85 cm^3^) tumor, was initially treated with two-staged ETCA and adjuvant PBRT. It was very difficult to reduce the whole tumor burden, and the tumor eventually recurred at the surgical bed near the cervicomedullary junction. Additionally, we performed a third ETCA to remove a mass near the medulla. An intradural drop metastasis was found again on a lumbar vertebra 62 months after the initial operation. We failed to control the tumor in another patient after 49 months. Patient 14 had undergone repetitive radiosurgery before admission to our institute. The neurovascular structures, particularly the ipsilateral third nerve and carotid artery, were severely encased and injured by previous radiation.

### 3.3. Progression-Free Survival (PFS)

#### 3.3.1. PFS Based on the Extent of Resection

GTR (100%) and NTR (>95%) were achieved in 35.3% (6 of 17 patients) and 52.9% (9 of 17 patients) of patients, respectively. We analyzed the extent of resection with regard to the overall outcome. The Kaplan–Meier survival curve revealed a significant benefit for the extent of the resection (Figure 1B). A log-rank analysis showed significant differences in the PFS rates between groups that achieved total resection and groups that did not (GTR vs. NTR: *p* = 0.002; GTR vs. STR: *p* < 0.001). The group in which tumor volume was not reduced enough (STR: <90%) had a significantly poorer PFS rate than that of a group in which a substantial amount of the mass was removed (GTR + NTR: >90%, *p* < 0.001). Two patients who achieved a limited STR showed 100% progression at 9 and 40 months after the first surgery, respectively. Tumor progression followed NTR in two (22.2%) patients and followed GTR in no patient (0%). The mean tumor volume (MTV) of the GTR group was 9.27 cm^3^, the MTV of the NTR group was 22.72 cm^3^, and the MTV of the STR group was 54.52 cm^3^. Larger tumors involving a greater area often required multiple surgeries and approaches to achieve radical resection.

#### 3.3.2. Progression-Free Survival Based on the Surgical Approach

We also divided the population into two groups based on the surgical approach used. The first group (group I) of 11 patients only underwent the ETCA, in which a large portion of the tumor was located in the midline. The other group (group II) included the remaining six patients who had tumors that had expanded laterally and required additional lateral skull-base approaches. Group I achieved about 90% NTR, and group II achieved 82% NTR. The incidences of progression were 9.1% and 33.3% in groups I and II, respectively. The probabilities of a residual brainstem lesion after resection were 18.8% and 33.3% in groups I and II, respectively. No postoperative cranial neuropathy was detected after the ETCA, but one patient (16.7%) in group II had permanent third-nerve palsy. The Kaplan–Meier curves did not show a difference in PFS between the two groups (Figure 1C). However, they supported the ETCA being able to accomplish radical resection as well as conventional skull-base surgeries, with both few complications and better functional outcomes. The 5-year PFS rate of patients who only underwent the ETCA was 87.5%, and that of patients who underwent a combined approach was 50.0%.

#### 3.3.3. Progression-Free Survival Based on a Remnant Brainstem Lesion

Figure 1D shows PFS rates based on whether or not a residual tumor remained at the surface of the brainstem (pons or medulla oblongata). Residual tumors were determined by the surgeon’s judgement and by postoperative MRI. These patients had masses at least 5 mm from the brainstem surface. This meant that marginal dose radiation therapy did not affect the normal brainstem. The Kaplan–Meier survival curve showed a significantly different actuarial PFS rate, regardless of surgical approach or adjuvant therapy (*p* = 0.023). The 5-year PFS rate for cases without a remnant brainstem lesion was 77.1%, and that of cases with a remaining brainstem lesion was 53.3%.

## 4. Discussion

Modern large patient series have revealed a 5-year clival chordoma survival rate of about 80% when no tumor is visible after surgery [10,23,24]. According to a study by Yoneoka et al. with the longest follow-up period (mean, 122 months; median, 108 months; 13 patients), the 5-year mean survival rate was 82.5% [3]. In our series, the 10-year OS and PFS rates were 94.1% and 67.4%, respectively. These results are comparable to previous studies and resulted from our multimodal treatment strategy rather than using a single operation for tumor control. Thus, a variety of radiation therapies and a diversity of surgical approaches should be considered to manage clival chordomas. Surgeons can now combine surgery with radiation therapy tailored to an individual. The physician must embrace several objectives, including a maximum safe surgical resection, long-term tumor control, and maintenance of good functional outcome.

Above all, the extent of resection is one of the most important factors when managing clival chordomas. Sen et al., who enrolled the greatest number of patients (71 patients) with clival chordomas, focused on complete resection [25]. The 5-year survival rate of patients who had radical resection was twice that of patients who did not. In addition, the tumor regrowth rate was 70% in cases of incomplete resection. Di Maio et al. conducted a 10-year meta-analysis and compared groups of patients who did and did not achieve complete resection. The 5-year risk of recurrence was 3.83-fold higher in patients with an incomplete resection than in those who had a complete resection, and the 5-year relative risk of mortality was 5.85-fold higher in patients who received an incomplete resection than in those with a complete resection [5]. Our results agreed with theirs and showed significantly different outcomes based on the extent of resection. Our 5-year PFS rate reached 53% unless we achieved GTR. In contrast, the tumor progression rate in the incomplete resection group was only 36.4% in our series. Because of the additive effect of early adjuvant radiation therapy, this value was lower than that reported in a previous study that only considered surgical aspects. Most progression occurred 3–5 years after the initial treatment; thus, we recommend a careful MRI study during that period.

The factors determining the resectability of a lesion in previous studies included mean tumor volume, the anatomical region involved with the tumor, previous surgery, and previous radiation [6,25,26,27]. Larger lesions are more infiltrative and destructive to adjacent structures, making them difficult to remove with only a single surgery. In our patients, MTV was larger and the anatomical areas involved were definably greater in the NTR and STR groups. We combined staged lateral skull-base surgery to radically remove such multiple compartmental lesions. A novice surgeon might leave a large amount of the mass for adjuvant radiation therapy. However, this is a critical issue. Varga et al. reported that patients receiving intralesional debulking experienced tumor progression in <2 years [28]. Thus, margin-free radical resection is mandatory. It cannot be emphasized enough that the goal of surgery is to remove the tumor completely, or as much as possible without morbidity.

Reoperations for recurrent disease are fraught with morbidity, particularly if the lesion surroundings were irradiated and scarred previously. Two of our patients who underwent GKRS had previous chronic neurological problems. One patient (patient 6) was referred to our hospital with quadriparesis caused by the regrowth of clival chordomas despite two GKRS procedures at another institute. The brainstem signal intensity had already changed on an MRI due to radiation. We immediately radically resected the tumor; however, the quadriparesis remained. In another case (patient 14), we performed the ETCA combined with the TCSA and carefully dissected adhesions and scar tissue in the intradural space. We encountered significant scarring of the oculomotor nerve resulting from prior surgery and radiation treatment.

In what follows, we retrospectively consider the surgical status for each of the strategy levels I~IV and discuss specific cases as examples.

Level I includes extradural chordomas with or without minimal dural involvement. Small extradural chordomas are usually contained in the sphenopetroclival synchondrosis, particularly in the rostral one-third of the clivus (Figure 2A–C). In our study, five patients (MTV: 0.84–5.76 cm^3^) were included in this group and all presented with diplopia. Dorello’s canal is the most commonly involved bony structure in patients with diplopia. Dorello’s canal is the entry point for the abducens nerve penetrating the clival dura. It is located 17–20 mm below the posterior clinoid process and is an important landmark to preserve the abducens nerve. Because Dorello’s canal is located behind the lateral edge of the dural window, the operator must extend the dural window by directly visualizing the abducens nerve with an angled endoscope.

The bone marrow should be radically removed while drilling the rostral clivus during the ETCA. Because these tumors occasionally only show an erosive lesion on the posterior surface of the clivus, a cranial base CT scan is necessary so that such a small lesion is not missed and to design the bone-drilling border. A marrow lesion is poorly enhanced using MRI but an irregularly lobulated erosive bony margin can be detected on a high-resolution CT scan (Figure 2D). A bone scan or positron emission tomography scan demonstrates an extracranial lesion for metastasized chordomas. However, demarcating a small skull-base chordoma lesion and migration of fluorodeoxyglucose to a skull-base lesion is difficult.

We evaluated histopathology intraoperatively at every corner of the lesion until we found normal bone tissue. The ETCA should be the initial approach for small extradural clival chordomas, as the dura is an important barrier against tumors. Although small tumors did not seem to invade the dura on perioperative images, we were always cautious when thinned or torn dura was found in the operative field. If a small defect in the dura is not repaired, the risk of tumor seeding into the intradural space or CSF leakage increases [29].

Level II is a moderate intradural invasion or a pure intradural mass. Clival chordomas in this group have a large portion of the mass occupying the prepontine cistern and compressing the brainstem. A parenchymal indentation of the anterior pontine surface causes poor brainstem function, dysarthria, disequilibrium, and quadriparesis. Five cases (MTV: 5.6–25.1 cm^3^) in this study required immediate decompression. The objective of the ETCA for level II surgery is to achieve a wider surgical window in the vertical and horizontal dimensions. The ETCA allows the operative field to be expanded to the entire ventral skull base, particularly the anterior and middle skull base. It is necessary to expose the sellar floor of the dura first to avoid blind drilling of clival bony structures. The surgeon could lose the anatomical orientation of the sphenoid sinus without a navigation system in cases of a presellar and conchal type of sphenoid sinus. If the sellar floor dura is exposed, it is easy to expose the clival dura by following their continuity. This procedure improved safety while resecting the clival bone directed toward the intradural space. We skeletonized the bilateral ICA to obtain a wider dural opening laterally to the paraclival area. The borders of the carotid arteries were identified using a navigation system and Doppler sonography. Most of the small tumors were extradural, whereas some of the larger tumors invaded the outer layer, extended through the inner layer, or invaded the intradural space [25]. It was difficult to visualize the venous bleeding caused by the basilar venous plexus being tangled with the tumor while exploring the prepontine space. The basilar artery, superior cerebellar arteries, and posterior cerebral arteries, as well as their perforators, should be preserved with meticulous dissection.

Tumors with anterosuperior protrusion invaded the suprasellar area and the cavernous sinus (Figure 3A–D). In these cases, patients can present with opthalmoplegia, facial pain, or pituitary dysfunctions. The tumor adhered to cranial nerves in a cavernous sinus and compressed the brainstem. We extended the bony resection superiorly, identified the sellar floor and dorsum sellae, and removed the posterior clinoid process posteriorly (Figure 3E). It was difficult to remove all of the calcified mass stuck in the cavernous sinus. After the surgery, the patient underwent GKRS for the remnant calcified mass in the cavernous sinus and showed no progression for 33 months (Figure 3F,G). Primary intradural chordomas without bony involvement are extremely rare, and only about 20 cases have been published [26,30,31,32,33,34]. Although an intradural chordoma seems to have no space between the parenchyma and the tumor, it has a rather good cleavage plane compared to that of an extradural lesion [33]. As the majority of primary intradural chordomas are located in the prepontine space, the ETCA was a feasible strategy for the case we encountered. We achieved NTR and it showed no progression for 52 months with adjuvant PBRT after the ETCA.

Level III is lateral extension. It was very difficult to achieve complete resection of tumors in this group. Above all, we focused on reducing the tumor burden as much as possible using the ETCA for midline lesions pressing on the brainstem, as mentioned above. All patients (MTV: 10.2–66.8 cm^3^) required additional lateral skull-base approaches because large portions of the tumors occupied Meckel’s cave, the petrous bone, and the cerebellopontine cistern. Selecting an approach for lesions lateral to the carotid arteries varies depending on the surgeon’s preference. Surgeons who prefer extended endonasal approaches extend the surgical window laterally using transpterygoid or transmaxillary approaches. CSF leakage represents a major problem when performing this type of expanded approach. The rate of CSF leakage in the endoscopic literature is as high as 30% [7,11,12,13,15,16]. CSF leaks can be hazardous and difficult to manage, and we do not overlook their impact on patients. Tumors intimately involved with the cavernous sinus or those with substantial extension to the brainstem require a secondary auxiliary approach to facilitate total tumor resection. Stippler et al. reported that conventional lateral skull-base approaches are very valuable. They believe that the ability to access skull-base tumors via corridors from all angles is critical [13]. In addition, lateral approaches have also evolved to be less destructive. We have tailored and minimized craniotomy to lesions. The extradural trans-cavernous sinus has an advantage in that a large amount of an extradural mass can be resected, and the intradural procedure has been minimized for lesions lateral to the upper clivus. A petrous apexectomy minimizes the destruction of bone but maximizes the visualization from Meckel’s cave to the cerebellopontine angle for lesions lateral to the middle clivus (Figure 4).

Level IV is a lower extension to the upper cervical area. One patient had a huge tumor (tumor volume: 98.75 cm^3^) that obstructed the nasopharynx (Figure 5). It had elongated from the entire clivus to the retropharyngeal soft tissue and to the tip of the C2 vertebral body downward. The posteriorly prominent mass pushed the lower medulla back deeply. We extended the ETCA surgical window to the lower clivus instead of using a transoral approach. We retracted the hard palate with a rubber band turned behind the velum and pulled back through the oral cavity to visualize the lower part of the clivus and the upper cervical level. The nasopharynx mucosa was opened onto the midline, and we reached the cervicomedullary junction after resecting the soft tissue lesions. Closing the mucosa with sutures permitted earlier healing and recovery, and no CSF leaks were detected. We performed a reoperation when the patient experienced tumor regrowth after 40 months. Radiation therapy is often used as an initial treatment because it is rare to safely achieve en bloc resection at the cervicomedullary junction or in the upper cervical region. However, repeated surgical debulking may be necessary for long-term tumor control, particularly in a healthy young patient [17].

We considered the role of the ETCA with adjuvant radiation therapy. The important surgical point is to secure the distance from the brainstem to the tumor. One problem for surgeons planning adjuvant radiation therapy is irradiating lesions stuck on critical structures, such as the brainstem or cranial nerves. If a tumor adheres to the brainstem, it is not only difficult to remove the mass but it is also difficult to irradiate it. When a proton beam is directly delivered to a remnant tumor near the brainstem, quadriparesis can occur as in our complication case (Appendix A). Substantial tumor debulking allows safe delivery of curative high-dose adjuvant radiotherapy. Thus, we focused on reducing the tumor burden in front of the brainstem as much as possible, even in cases of an unresectable mass (Figure 2). Significant relationships exist among the tumor–brain contact interface area, tumor volume, and radiation injury after stereotactic radiosurgical procedures [35,36]. Because clival chordomas have an epicenter of tumor growth in the extradural space, ventral decompression is more effective to lighten the burden and secure the brainstem–chordoma interface [7,12,15]. The ETCA yields wide, full-length visualization of the extradural epicenter and leaves less of a brainstem lesion remnant postoperatively. Thus, the more we relieved the brainstem–chordoma interface, the higher the dose of local irradiation that was possible. Resections with negative margins can be technically difficult, and even GTR may not translate to oncologically microscopic negative margins. Although maximum and radical resection is mandatory, planned NTR or STR is the optimum treatment for neurological preservation followed by radiation therapy [6]. This strategy has already been employed in previous series with acceptable survival and complication rates. Because radiation therapeutic technology continues to develop, various adjuvant modalities are essential to control clival chordomas [18,19,20,21,22]. The endoscopic approach is considered a strategic minimally invasive tool. The ETCA yields tumor resection rates at least equivalent to those of conventional skull-base approaches and with less morbidity. With the ETCA, tumors could be sharply dissected off from the cranial nerves and brainstem. Consequently, this strategy reduces tumor burden in close proximity to vital structures and results in better functional outcomes from adjuvant radiotherapy.

We also report extraordinary cases of rapidly growing or distant metastasis. Some chordomas have a mild evolution and grow slowly, whereas others have more aggressive behavior, with rapid local recurrence and distal spreading. In our study, patient 16 demonstrated extraordinarily rapid progression within several months. However, there are a lack of markers to indicate the aggressiveness level of a chordoma. Pallini et al. reported that mRNA expression of human telomerase reverse transcriptase (hTERT) is a predictor of rapid growth and the recurrence rate of chordomas [2]. If a tumor was partially resected, a p53 protein mutation and hTERT mRNA expression predicted an increased doubling time for the residual tumor as well as the probability of tumor recurrence. Ito et al. found that the MIB-1 labeling index is independently associated with recurrence [4]. They suggested that the optimum cutoff point for the MIB-1 labeling index was 3.44%. New treatment modalities may be required in the future, such as chemotherapy and molecular- or gene-targeted therapies. Our patient 17 presented with a distal metastasis on a lumbar vertebra 62 months after initial treatment and 22 months after the last operation. Fewer than 10 intradural drop metastasis cases have been reported [37,38]. It is worth considering various routes of surgical spread for chordomas and intradural, extradural, and local metastases. Al-Mefty et al. coated the walls of the operative tunnel with fibrin glue and large cotton patties and carefully removed them at the end of surgery [38]. We also recommend keeping gelatin sponges in the dependent portion of the operative field as a dam or barrier. Spinal MRI is appropriate for patients with intracranial chordomas and provides earlier detection of metastasis or multiple synchronous chordomas [37,39].

There were some limitations in our study. We conducted this study to highlight the advantages of endoscopic surgery and not to exclude conventional skull-base approaches. Our results do not mean that the ETCA is superior to other conventional approaches. However, the ETCA has acquired a reputation in the field of skull-base surgery as a minimally invasive technique. Furthermore, radiation therapy technology has made remarkable progress; thus, such adjuvant modalities support using less invasive surgical approaches. In fact, radiation therapy played a considerable role in improving the overall outcomes in this study. We did not analyze radiation therapy because we lack the relevant data in the surgical department. In addition, we lacked knowledge about the pathological conditions of each case. Future studies should comprehensively consider such factors, and there is the need for a more detailed statistical analysis to quantify the impact of tumor size on progression-free survival (PFS) and overall survival (OS). Furthermore, we will clarify our surgical techniques and outcomes, address discrepancies in the surgical strategy levels, and provide more information on complications such as cerebrospinal fluid leaks in a future study. For clear reporting, we showed our data in Table 1. We also understand the importance of including a discussion on cranial nerve improvement rates in future studies. Overall, a larger and more homogeneous patient sample is needed to draw more robust conclusions in any future study.

## 5. Conclusions

The endonasal transclival approach (ETCA) offers a significant extent of resection for clival chordomas based on tailored surgical considerations. Access to the clival area necessitates a mix of extended lateral, posterolateral, and anterior skull-base approaches. This study underlines the significance of tailoring surgical considerations based on the individual patient’s condition. Given the advances in skull-base approaches and radiation therapy, a multimodal strategy is recommended for effective chordoma management. There remains a necessity for more defined future studies, especially those considering different groups based on the extent of tumor spread, to further enhance our understanding and treatment strategies for clival chordomas.

## Figures and Tables

**Figure 1 jcm-12-04460-f001:**
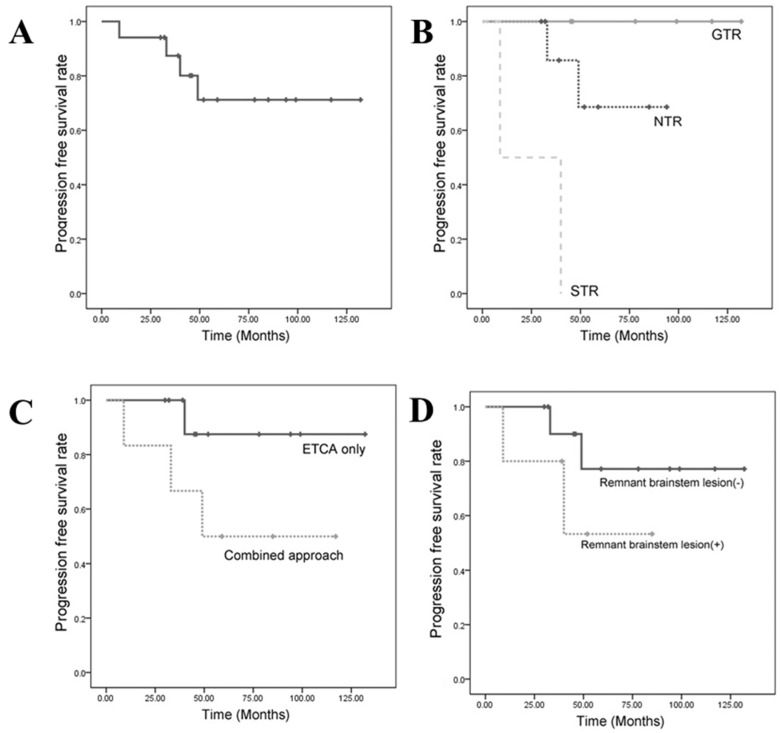
Kaplan–Meier curves for PFS (progression-free survival) rates in patients with clival chordomas. (**A**) Overall PFS rates in all 17 patients. (**B**) PFS rates in patients with GTR (100%), NTR (>90%), and STR (50–90%). PFS rates differed significantly among the three groups by log-rank test (*p* < 0.001). (**C**) PFS rates in patients operated on using ETCA and combined approaches. (**D**) PFS rates in patients with and without postoperative remnant brainstem lesion. PFS rates differed significantly between the two groups by log-rank test (*p* = 0.001).

**Figure 2 jcm-12-04460-f002:**
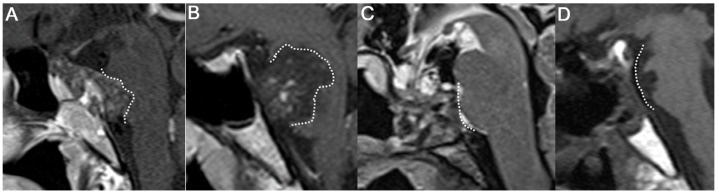
Extradural chordomas with or without minimal dural involvement. (**A**,**C**) A typical small extradural chordoma occupying upper clivus and minimal dural contact. (**B**) A chordoma contained in the clivus with an erosive lesion. (**D**) Postoperative sagittal CT showing bony resection margin from sella to middle of clivus. (C: clivus P: pons, Tm: tumor).

**Figure 3 jcm-12-04460-f003:**
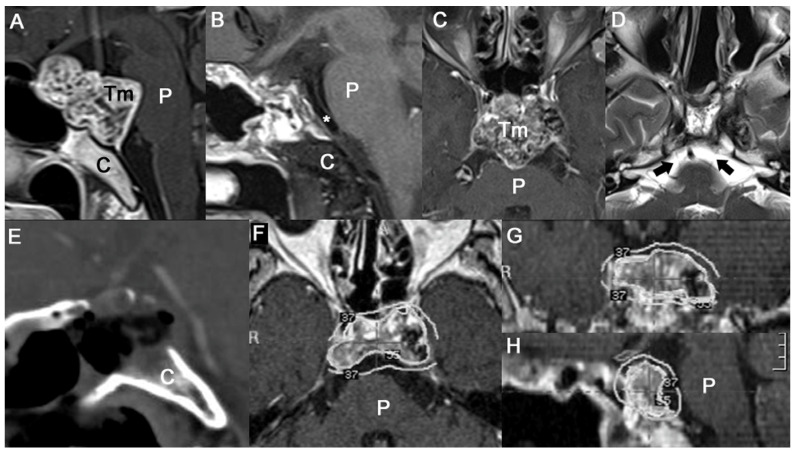
Illustrative case of Level II strategy. (**A**) Preoperative exophytic mass compressing upper and middle pons. (**B**) Postoperative images showing released whole anterior surface of pons and released prepontine cisterns with basilar artery (asterix). (**C**) Preoperative mass in upper pons occupying cavernous sinus and bilateral parasellar area. (**D**) Postoperatively, bilateral abducens nerves (black arrows) were released at middle and lower prepontine cistern. (**E**) Postoperative sagittal CT showing bony resection margin of entire sellar floor and upper half of clivus. (**F**–**H**) Adjuvant GKRS was done at intracavernous lesions but sparing chordoma–brainstem contact interface. (C: clivus; P: pons; Tm: tumor; R: right side, *: basilar artery).

**Figure 4 jcm-12-04460-f004:**
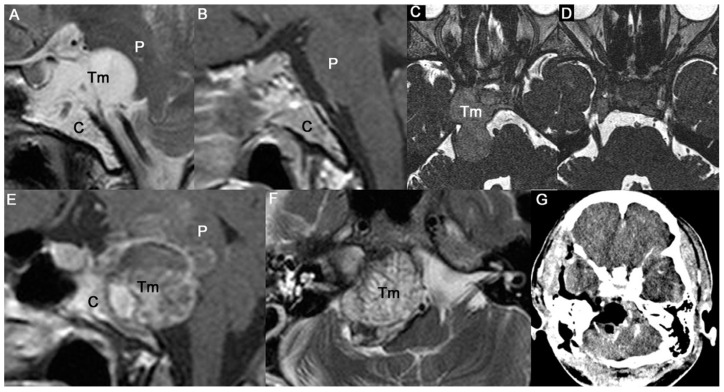
Illustrative cases of Level III strategy. (**A**,**C**) Preoperative MR images shows laterally expanded clival chordomas occupying petrous apex, Meckel’s cave, and upper cerebellopontine cistern. (**B**) Postoperative MR images showing midline clival area removed after ETCA. (**D**) Postoperative MR images showing released ipsilateral trigeminal nerve and cerebellopotine angle by the right side TCSA. (**E**,**F**) Preoperative MR images showing large chordoma occupying middle and lower clivus with severely distorted cerebellopontine area. (**G**) Postoperative axial CT scan showing tailored surgical corridor at petrous apex. (C: clivus; P: pons; Tm: tumor).

**Figure 5 jcm-12-04460-f005:**
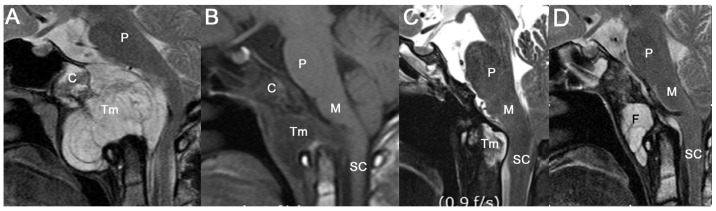
Illustrative case of Level IV strategy. (**A**,**B**) Preoperative sagittal MR images showing clival chordoma extensively occupying nasopharynx, and elongated from the lower pons to cervicomedullar area. At first, it looks hard to remove completely in one stage surgery. (**C**) Remnant tumor after initial ETCA that showed progression at 40 months after PBRT. (**D**) Final postoperative MR images showing fat graft inserted at surgical bed and cervicomedullary junction relieved after reoperation (ETCA). Level IV need combined treatment strategy in overall. (C: clivus; F: fat graft; M: medulla oblongata; P: pons; SC: spinal cord; Tm: tumor).

**Table 1 jcm-12-04460-t001:** Summary and demographics of present study cases.

Case	Age /Sex	F/U (Mos)	MTV (cm^3^)	Level	Preoperative Status	Strategy	EOR	Postoperative Status	Adjuvant RTx	Disease Progression
1	49/F	46	1.76	I	Diplopia	ETCA	GTR	No symptom		Stable
2	37/F	45	0.84	I	Diplopia	ETCA	GTR	No symptom		Stable
3	59/F	132	1.00	I	Diplopia	ETCA	GTR	No symptom		Stable
4	49/F	30	5.65	I	Headache, regrowth after TSA	ETCA	STR	No symptom Transient DI (resolved)	PBRT	Stable
5	12/F	78	5.34	II	Headache	ETCA	GTR	No symptom		Stable
6	47/F	36	5.67	II	Regrowth after previous GKRS quadriparesis	ETCA	GTR	Remained quadriparesis		Stable
7	4/M	94	5.76	II	Diplopia	ETCA	NTR	No symptom	PBRT	Stable
8	43/M	32	25.1	II	Diplopia	ETCA	NTR	No symptom	GKRS	Stable
9	32/M	52	22.95	II	Dysequilibrium	ETCA	NTR	No symptom CSF leak (resolved)	PBRT	Stable
10	42/F	39	66.8	III	Regrowth after previous surgery	ETCA	NTR	No symptom Transient facial palsy (resolved)	PBRT	Stable
11	57/F	117	26.4	III	Trigeminal neuralgia	ETCA + TCSA	GTR	No symptom		Stable
12	48/F	59	10.1	III	Diplopia	ETCA + TCSA	NTR	No symptom	PBRT	Stable
13	51/F	85	48.7	III	Diplopia	ETCA + APTA	NTR	No symptom	PBRT	Stable
14	48/M	61	9.37	III	Regrowth after previous GKRS ophthalmoplegia	ETCA + TCSA	NTR	Remained ophthalmoplegia	GKRS	Regrowth at 49 months → Reoperation
15	34/F	117	10.2	III	Diplopia, trigeminal neuralgia	ATPA	NTR	No symptom	GKRS	Regrowth at 33 months → GKRS
16	35/F	9	10.3	III	Altered mental status	ATPA	STR	Unchanged		Death at 9 months
17	13/F	62	98.75	IV	Dysphagia, dysarthria	ETCA	STR	No symptom	GKRS	Regrowth at 40 months → Reoperation + GKRS

Level I: Extradural with or without minimal dural contact; Level II: moderated intradural invasion or pure intradural mass; Level III: lateral extension; Level IV: lower extension to upper cervical area; MTV: mean tumor volume; EOR: extent of resection; ETCA: endonasal transclival approach; TCSA: trans-cavernous sinus (Dolenc’s) approach; ATPA: anterior transpetrosal approach; TSA: transsphenoidal approach; GTR: gross total resection; NTR: near total resection (>90%); STR: subtotal resection (50–90%); RTx: radiation therapy; PBRT: proton beam radiation therapy; GKRS: Gamma Knife radiosurgery.

## Data Availability

The data is unavailable due to ethical restrictions.

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
