# Peer review of "Long-Term Outcomes after Multimodal Treatment for Clival Chordoma: Efficacy of the Endonasal Transclival Approach with Early Adjuvant Radiation Therapy"

_jcm, 2023, doi:10.3390/jcm12134460_

Round 1
Reviewer 1 Report
The authors describe extensively their experience in treatment of clival chordoma during a period of 11 years (2002-2013). The number is not very high (17 patients) however the mean follow-up is the longest published taking into account all the treatment performed (66 months).
The article is interesting because describe the serie of one hospital where all kind of treatment is used for these rare tumors.
The conclussion does not provide any new novelty in the treatment of chordoma, but the experience and description of each type of chordoma is still interesting to read.
Author Response
Dear Reviewer 1,
Thank you for your time and thoughtful comments on our manuscript titled "Long-term outcomes after multimodal treatment for clival chordoma: Efficacy of the endonasal transclival approach with early adjuvant radiation therapy." We truly appreciate your insightful feedback, which has significantly improved our paper.
You noted our comprehensive discussion on our experience treating clival chordomas over an 11-year period, as well as the extensive description of each type of chordoma, which you found interesting. We are glad you recognized the strength of our study, particularly the long mean follow-up period which is indeed one of the longest published.
Regarding the sample size, we agree that it is relatively small. However, as you've noted, clival chordomas are a rare type of tumor and given the rarity of the condition, our sample size is substantial in this context.
You pointed out that our conclusion does not offer any novelty in the treatment of chordomas. Our main intent with this paper was to provide a detailed description of our experiences and observations over an extended period rather than to introduce a completely new treatment. We believe that the cumulative knowledge and insights from detailed experiences like ours will eventually contribute to innovative treatments for these types of tumors. We have revised our conclusion to more clearly reflect this intention and the implications of our findings for practice as below sentence with yellow line and square [ ]. We deeply appreciate your suggestions and constructive feedback which have greatly helped in refining our manuscript.
- The Endonasal Transclival Approach (ETCA) offers a significant extent of resection for clival chordomas based on tailored surgical considerations. Access to the clival area necessitates a mix of extended lateral, posterolateral, and anterior skull-base approaches. [This study underlines the significance of tailoring surgical considerations based on the individual patient's condition.] Given the advancements in skull-base approaches and radiation therapy, a multimodal strategy is recommended for effective chordoma man-agement. [There remains a necessity for more defined future studies, especially those considering various groups based on the extent of tumor spread, to further enhance our understanding and treatment strategies for clival chordomas.]

Reviewer 2 Report
(1)It would be helpful to define "combined approaches" in the abstract, as comparing this to ETCA is the point of this paper. It is also not clear how the ETCA "enables" radiotherapy; perhaps the authors imply that debulking makes the radiotherapy safer as implied by lines 480-484? Clarification would be useful.
(2) The English is in need of much improvement. For example, Lines 35-36 are difficult to understand, likely due to some language issue...similarly, in Line 42, should be "recurrent" not "recurred," and many other such issues. Please have this manuscript reviewed and edited by a native-level English speaker
(3)Most importantly, this is a single center, retrospective, non-controlled review of outcomes assessed by the treating physicians for a small number of patients (recognizing that large series are difficult with an uncommon tumor). Without a more objective comparison that clarifies the role of patient selection and many other such factors, it is difficult to conclude more than that the apparent good results justify a true randomized, controlled trial. Dividing the patients into a group with midline tumors amenable to ETCA and a group with larger tumors requiring combined approaches, and then concluding that the first group did well, proves only that patients with worse disease have worse outcomes. To demonstrate superiority or even non-inferiority of the ETCA approach to other treatment options would require randomization of comparable patients and objective evaluation of outcomes.
See my comments above. There are numerous grammatical errors and incomplete or difficult-to-interpret sentences.
Author Response
Dear Reviewer 2,
Thank you for your time and thoughtful comments on our manuscript titled "Long-term outcomes after multimodal treatment for clival chordoma: Efficacy of the endonasal transclival approach with early adjuvant radiation therapy." We deeply appreciate your insightful review comments, which have significantly improved our paper.
(1)It would be helpful to define "combined approaches" in the abstract, as comparing this to ETCA is the point of this paper. It is also not clear how the ETCA "enables" radiotherapy; perhaps the authors imply that debulking makes the radiotherapy safer as implied by lines 480-484? Clarification would be useful.
à Thank you for wonderful comments. As the surgical experience accrues, the scope of the Endonasal Transclival Approach (ETCA) for clival chordomas treatment is gradually extending towards the lateral side. However, for junior neurosurgeons or when tumors have extended laterally beyond the petroclival fissure, the implementation of a lateral skull base approach becomes necessary.
The term 'combined approaches,' a central aspect of our study, refers to a mix of techniques utilized based on the location of the lateral invasion. These approaches include, but are not limited to, the extradural transcavernous approach, the far lateral transcondylar approach [Fava, Arianna, et al.] and the anterior petrosal approach[Kim, Seong Min, et al.]. Each technique provides distinct advantages depending on the tumor's specific characteristics and spread, and plays a critical role in managing clival chordomas. Therefore, we suggest these combined approach for proper management. The authors define "combined approaches" and add paragraph on introduction line 34-37 as below
- The combined approaches refers to a mix of techniques utilized based on the location of the lateral invasion, including, but are not limited to, the extradural transcavernous approach, the far lateral transcondylar approach and the anterior petrosal approach.
And more clarify that tumor location and maximal debulking enables radiotherapy more safe on manuscript lines 480-484 with supplementary figure 1 as below.
- We consider the role of the ETCA with adjuvant radiation therapy. The important 480 surgical point is to secure the distance from the brainstem to the tumor. One problem for 481 surgeons planning adjuvant radiation therapy is irradiating lesions stuck on critical struc- 482 tures, such as the brainstem or cranial nerves. If a tumor adheres to the brainstem, it is not 483 only difficult to remove the mass but it is also difficult to irradiate it. [When proton beam is directly transferred remnant tumor near brain stem, quadriparesis can be occurred as our complication case (supplementary figure 1)]
(2) The English is in need of much improvement. For example, Lines 35-36 are difficult to understand, likely due to some language issue...similarly, in Line 42, should be "recurrent" not "recurred," and many other such issues. Please have this manuscript reviewed and edited by a native-level English speaker
à Thank you for valuable comments. The authors revise the paragraph as below, and also perform additional native-level English correction and attach proof of English correction if needed.
The endonasal transclival approach (ETCA) was tailored to the localization and craniocaudal extension of the clival chordoma based on central skull base
à The Endonasal Transclival Approach (ETCA) was applied according to the specific location and the craniocaudal spread of the clival chordoma to the central skull base.
Adjuvant radiotherapy for residual or recurred tumors is also recommended for long-term tumor control
à Adjuvant radiotherapy for residual or recurrent tumors is also recommended for long-term tumor control
à Also a few grammatical errors were revised at this revision stage, and more English correction will be performed with native English speaker additionally.
(3)Most importantly, this is a single center, retrospective, non-controlled review of outcomes assessed by the treating physicians for a small number of patients (recognizing that large series are difficult with an uncommon tumor). Without a more objective comparison that clarifies the role of patient selection and many other such factors, it is difficult to conclude more than that the apparent good results justify a true randomized, controlled trial.
Dividing the patients into a group with midline tumors amenable to ETCA and a group with larger tumors requiring combined approaches, and then concluding that the first group did well, proves only that patients with worse disease have worse outcomes. To demonstrate superiority or even non-inferiority of the ETCA approach to other treatment options would require randomization of comparable patients and objective evaluation of outcomes.
à Thank you for excellent comments. Regarding the sample size, we agree that it is relatively small. And this is retrospective observational analysis, some limitations were inevitable. If our study group gather more patients for analysis, we can also perform propensity score matched analysis consider tumor size or location in future study. As you've noted, clival chordomas are a rare type of tumor and given the rarity of the condition, our sample size is our best effort in this context.
You pointed out that our study does not offer any superiority or even non-inferiority of the ETCA approach to other treatment options in the treatment of chordomas.
Our main intent with this paper was to provide a detailed description of our experiences and observations over an extended period rather than to introduce a completely new treatment. We believe that the cumulative knowledge and insights from detailed experiences like ours will eventually contribute to innovative treatments for these types of tumors. We have revised our conclusion to more reflect this intention based on your valuable comments and the implications of our findings for practice as below sentence with yellow line and square [ ]. We deeply appreciate your suggestions and constructive feedback which have greatly helped in refining our manuscript.
- The Endonasal Transclival Approach (ETCA) offers a significant extent of resection for clival chordomas based on tailored surgical considerations. Access to the clival area necessitates a mix of extended lateral, posterolateral, and anterior skull-base approaches. [This study underlines the significance of tailoring surgical considerations based on the individual patient's condition.] Given the advancements in skull-base approaches and radiation therapy, a multimodal strategy is recommended for effective chordoma man-agement. [There remains a necessity for more defined future studies, especially those considering various groups based on the extent of tumor spread, to further enhance our understanding and treatment strategies for clival chordomas.]
References
Fava, Arianna, et al. "Endoscope-assisted far-lateral transcondylar approach for craniocervical junction chordomas: a retrospective case series and cadaveric dissection." Journal of Neurosurgery 135.5 (2021): 1335-1346.
Kim, Seong Min, et al. "Cochlear line: a novel landmark for hearing preservation using the anterior petrosal approach." Journal of Neurosurgery 123.1 (2015): 9-13.

Round 2
Reviewer 2 Report
Thank you for taking the time to consider my concerns and make some modifications. I think these changes contextualize the report much better and clarify its intent.